# High prevalence of gastroschisis in Brazilian triple side border: A socioenvironmental spatial analysis

**Suzana de Souza**[1]*, **Oscar Kenji Nihei**[2], **Cezar Rangel Pestana**[1]

**1** Federal University for Latin-American Integration, Latin-American Institute for Life and Nature Sciences, Foz do Iguassu, Paraná, Brazil, **2** State University of West Paraná, Foz do Iguassu, Paraná, Brazil

* suzanaesouza@gmail.com

**Data Availability Statement:** All relevant data are within the manuscript and its Supporting Information files.

**Funding:** The authors received no specific funding for this work.

## Abstract

This research investigated the spatial association between socioenvironmental factors and gastroschisis in Brazilian triple side border. A geographic analysis for gastroschisis prevalence was performed considering census sector units using Global Moran Index, Local Indicator of Spatial Association Analysis and Getis Ord statistics. Sociodemographic factors included rate of adolescent and parturients over 35 years; population with no income and above 5 minimum wages; rate of late prenatal; and proximity to power transmission lines. Logistic regression models were applied to verify the association between socio-environmental factors and prevalence of gastroschisis. No global spatial correlation was observed in the distribution of gastroschisis (Moran´s I = 0.006; p = 0.319). However, multiple logistic regression showed census sectors with positive cases had higher probability to power transmission lines proximity (OR 3,47; CI 95% 1,11–10,79; p = 0,031). Yet, spatial scan statistic showed low risk for gastroschisis in southern city region (OR = 0; p = 0.035) in opposite to power transmission lines location. The study design does not allow us to attest the causality between power transmission lines and gastroschisis but these findings support the potential exposure risk of pregnant to electromagnetic fields.

## Background

Gastroschisis is a birth defect characterized by abnormal abdominal wall closure with externalization of intra-abdominal structures. The defect is located in the paraumbilical region most common in the right side [1]. Cases of gastroschisis have increase worldwide from 1/50,000 live births up to 20-fold in recent decades [2]. Reduced maternal age is the only risk factor but an increase in all age groups has also been observed. The prevalence of gastroschisis may vary according to socioeconomic status, race, access to health services, nutrition, lifestyle and maternal education [3]. These determining factors can be addressed by preconception health care and early diagnosis.

Power Transmission Lines (PTL) emit non-ionizing radiation and its energy intensity is considered unable to break nucleic acids bonds [4, 5]. However, studies demonstrate

**Competing interests:** The authors have declared that no competing interests exist.

electromagnetic fields EMF can penetrate cells and interact with biomolecules [6, 7]. An association between maternal residence close to PTL and abortion, birth defects and prematurity in newborns is also reported [8–10].

Foz do Iguassu hosts Itaipu Binacional as one of the largest hydroelectric power stattion in the world. The 50 Hz energy flow uses a direct current system, while the 60 Hz energy flow uses a 765 kV system. The PTL transmits the energy produced to other locations by crossing inhabited city areas. However, there are no studies addressing the potential impact of this proximity on population health in these exposure areas.

The Prevalence Rate of Gastroschisis (PRG) in Foz do Iguassu has been 6.93/10,000 live births in recent years [11], higher than average of 3/10,000 live births [12–15]. The aim of this research was to investigate the potential spatial association between socioenvironmental factors and gastroschisis in Foz do Iguassu. This knowledge can contribute to both urban planning and preventive health care.

# Materials and methods

## Study design, setting and population

Ecological study with spatial approach included census sectors of Foz do Iguassu as the observation unit. The city is located in Brazil-Paraguay-Argentina triple border with 263,915 habitants distributed in 327 census sectors including 320 urban and 7 rural areas [16]. Spatial correlation analysis considered census sectors of urban area. Hydroelectric and the substation responsible to transmit power are located in the northern part of the city. Population was composed of all local live births in the period from 2012 to 2017. Records with "anomaly identification" empty or with a code for "Ignored" were excluded.

## Data sources and study variables

The study used Information System on Live Births (*Sistema de Informação Sobre Nascidos Vivos*—SINASC) and Brazilian Institute of Geography and Statistics (IBGE) - 2010 demographic census as information sources [16]. SINASC is a birth database with records extracted from Declaration of Live Birth form. Data were requested to Municipal Health Department and exported to Microsoft® Excel® spreadsheets.

The variables obtained from SINASC were: Type of congenital anomaly (Gastroschisis (Q79.3, according to International Classification of Diseases–ICD)); Parturients age (Presented at SINASC as a continuous quantitative variable; in this research was categorized as adolescent (up to 19 years old), adult (20 to 34 years old) and advanced age (over 35 years old)); Prenatal start period (Presented at SINASC as a continuous quantitative variable; in this research was categorized as early prenatal care (beginning in the first semester of pregnancy) and late prenatal care (beginning after the first semester of pregnancy)). The variables obtained from IBGE were related to the "residents *per capita* income" (no income and above 5 minimum wages).

## Power transmission lines

PTL were distributed according to energy towers latitude and longitude data using Google Earth™ version 7.15 software. All points were georeferenced on shapefile maps with SIRGAS2000 projection.

## Data analysis

Latitude and longitude birth addresses were obtained using the Bathgeo web resource. Spatial analysis was performed based on gastroschisis cases in each census sector. To perform this calculation, all newborns with and without gastroschisis were georeferenced on the shapefile maps with SIRGAS2000 projection and counted according to the census sector. The gastroschisis prevalence was calculated according to the following formula:

$$PRG = \frac{Number\ of\ gastroschisis\ cases\ in\ the\ census\ sector}{Number\ of\ live\ births\ in\ the\ census\ sector} \times 1,000$$

The exploratory spatial analysis of PRG was performed applying the Global Moran Index (Moran´s I), Local Indicator of Spatial Association (LISA) Analysis and Getis Ord (G) statistics using the Geoda$^{TM}$ software, version 1.12.1.131.

## Global Moran´s Index

Univariate Global Moran´s Index (Moran´s I) is a test with null hypothesis for spatial independence; in this case, its value would be zero. Positive values (between 0 and +1) indicate direct correlation and negative values (between 0 and -1) inverse correlation. The Moran´s I provides a single measure for all census sectors [17]. Moran´s I is expressed by the following formula:

$$I = \frac{\sum_i \sum_j W_{ij} Z_i.Z_j / S_o}{\sum_i z_i^2 / n}$$

Univariate Moran scatter plot consists of a spatially lagged variable on the y-axis and the original variable on the x-axis. Slope of the linear fits to scatter plot equals Moran's I. In Moran´s I analysis, queen configuration was utilized as continuity weight and horizontal, vertical and diagonal neighbor census sectors were considered.

Scatter plot was decomposed in four quadrants. Upper-right and lower-left quadrants correspond to positive spatial autocorrelation (similar values at neighboring locations). We refer to them as high-high and low-low spatial autocorrelation respectively. In contrast, lower-right and upper-left quadrants correspond to negative spatial autocorrelation (dissimilar values at neighboring locations). We refer to them as high-low and low-high spatial autocorrelation respectively.

## Local Indicator of Spatial Association analysis (LISA)

LISA was applied to local spatial association analysis to produce a specific value for each census sector and identify local spatial clusters with high or low prevalence [18]. The local version of Moran´s Index for each region i and year t is written as:

$$I_{i,t} = \frac{(x_{i,t} - \mu_t)}{m_o} \sum_j W_{ij} \left( x_{j,t} - \mu_t \right) \ \ with \ \ m_o = \sum_i \left( x_{i,t} - \mu_t \right)^2 / n$$

Where $x_{i,t}$ is the observation in region $i$ and year $t$, $\mu t$ is the average among regions in year $t$ and the sum over $j$ is those with only neighboring values included [19]. LISA provides a statistically significant degree of spatial autocorrelation in each spatial unit. The cluster map obtained by Moran dispersion diagram and LISA statistics combination allows a more adequate geographic visualization of degree of concentration of the studied variable [20]. In LISA analysis, queen configuration was utilized as continuity weight.

## Getis-Ord Statistics

Getis-Ord Statistics (G) is also a measure of local spatial association. This statistic for each region $i$ and year $t$ can be written as follows:

$$G_{i,t}(d) = \sum j \neq i \, x_{ij}(d) x_{j,t} / \sum j \neq i \, x_{j,t}$$

Where $x_{ij}$ ($d$) are elements of a symmetric binary space weight matrix equal to one for all links within the distance $d$ of a given region $i$ is equal to zero for all other links, including the region $i$ link to yourself [19].

The interpretation of G statistics is straightforward: a value larger than the mean (or, a positive value for a standardized z-value) suggests a high-high cluster or hot spot, a value smaller than the mean (or, negative for a z-value) indicates a low-low cluster or cold spot. In Getis-Ord Statistics, the queen configuration was utilized as continuity weight.

## Spatial scan statistic

Spatial scan statistic technique was developed by Kulldorff and Nagarwalla (1995). The search for risk groups is performed by positioning a virtual circle of variable radius around each centroid and calculating the occurrence rate in each virtual circle. If the observed value of the region limited by the circle is larger than expected, it is called a risk cluster; if the value is lower than expected, it is called a low-risk or protective cluster, with this procedure being repeated until all centroids are tested [21–23]. For identification of risk clusters for gastroschisis, Poisson discrete model was used considering the number of events in each area distributed according to a known risk. Null hypothesis is number of cases expected in each area is proportional to the size of its population. In Poisson model, the scanning statistic adjusts irregular population densities and analyzes the total number of cases observed [24]. The standard configuration applied by SaTScan<sup>TM</sup> software adopted the following criteria: no geographic overlap of the clusters, maximum cluster size equal to 50% of exposed population, circular-shaped clusters and 999 replications. Analyzes were purely spatial variation with Relative Risk (RR) and p values. RR refers to analysis a risk outcome within a geographically limited region, such as a census sector, defined as the risk $\lambda_Z$ in the region compared to risk in all other regions [21, 25]:

$$\lambda_Z = \frac{E(Y_Z)}{E_Z},$$

$$E_Z = N \frac{P_Z}{P_+},$$

where $Y_Z$ is Poisson random variable of Z-region count, with expected number given by $E(Y_Z)$; $P_Z$ is the population of Z region; $P_+$ E(Yz); PZ is the population of Z region; P+ is the total population at risk in an area; and N is the total number of cases. In the same way, $\lambda_A \setminus Z$ was also defined. Thus, the true relative risk is given as [21, 25]:

$$RR = \frac{\lambda_Z}{\lambda_{A/Z}}.$$

If both Z and A\Z have the same λZ = λAZ = λ, the relative risk is 1. Assuming that Z is selected independent of the observed values, the estimated relative risk is given by [21, 25]:

$$\hat{RR} = \frac{N_Z/E_Z}{(N - N_Z)/(E_A - E_Z)}$$

where N is the total number of cases, $N_Z$ is the number of cases in Z cluster; $E_A$ is the number of expected cases in the region under null hypothesis; $E_Z$ is the number of cases in the Z area under the null hypothesis. For interpretation, when RR is equivalent to 1, there is strong evidence to no cluster of risks on the map; if RR is below to 1, there is low risk or the area is protected; and above 1 represents a risk area. The program used for the spatial scan statistic was SaTScan™ version 9.6.

## Logistic regression

Simple and multiple logistic regression models were applied to verify the association between sociodemographic variables, maternal residence close to PTL and gastroschisis prevalence. Logistic regression was chosen due to non-parametric distribution of data. Gastroschisis was considered as a dependent variable and the rate of adolescent parturients (RAP), rate of advanced age parturient (RAAP), population without income, population with income above 5 minimum wages, late prenatal rate and proximity to PTL were considered as independent variables.

To calculate the RAP, parturients under the age of 19 were geo-referenced according to census sector. The rate was calculated according to the following formula:

$$RAP = \frac{Number\ parturients\ under\ the\ age\ of\ 19\ in\ the\ census\ sector}{Number\ of\ live\ births\ in\ the\ census\ sector} \times 1,000$$

To calculate RAAP, parturients over the age of 35 were geo-referenced according to census sector. The rate was calculated according to the following formula:

$$RAAP = \frac{Number\ parturients\ over\ 35\ years\ old\ in\ the\ census\ sector}{Number\ of\ live\ births\ in\ the\ census\ sector} \times 1,000$$

To calculate the rate of late prenatal care (RLPC), mothers who started prenatal care from 4th month of pregnancy were selected and georeferenced according to census sector. The rate was calculated according to the following formula:

$$RLPC = \frac{Number\ parturients\ with\ late\ onset\ of\ prenatal\ care\ in\ the\ census\ sector}{Number\ of\ live\ births\ in\ the\ census\ sector} \times 100$$

The variable "without income" and "income above 5 minimum wages" is expressed in absolute numbers according to the number of households in census sector. The dependent variable was dichotomized into 0 and 1, where 0 are census sectors without cases of gastroschisis and 1 are census sectors with cases of gastroschisis. For the independent variable proximity to PTL, a distance matrix was created between the centroid of each census sector and a point closer to PTL. After empirical tests, the distance that included all census sector close to PTL was 850 meters. Census sector whose centroid was less than 850 meters from any point in PTL was considered exposed with the value 1, whereas census sector with centroid more than 850 meters from PTL was considered not exposed with a value of 0. Other variables were dichotomized based on their median, census sectors with independent variable above were considered exposed and below considered not exposed. First, a simple logistic regression analysis was performed for all independent variables. Statistical value of $p \leq 0.20$ were included in a multiple logistic regression model for both models. Odds Ratio (OR) was calculated with 95% Confidence Intervals (95% CI). The software used in the logistic regression analysis was EpiInfo ™ version 7.2.

## Ethics review

This research was approved by Ethics Review Board of the Universidade Dinâmica das Cataratas; evaluation number: 2,856,426; Certified Ethical Presentation number: 92477918.0.0000.8527.

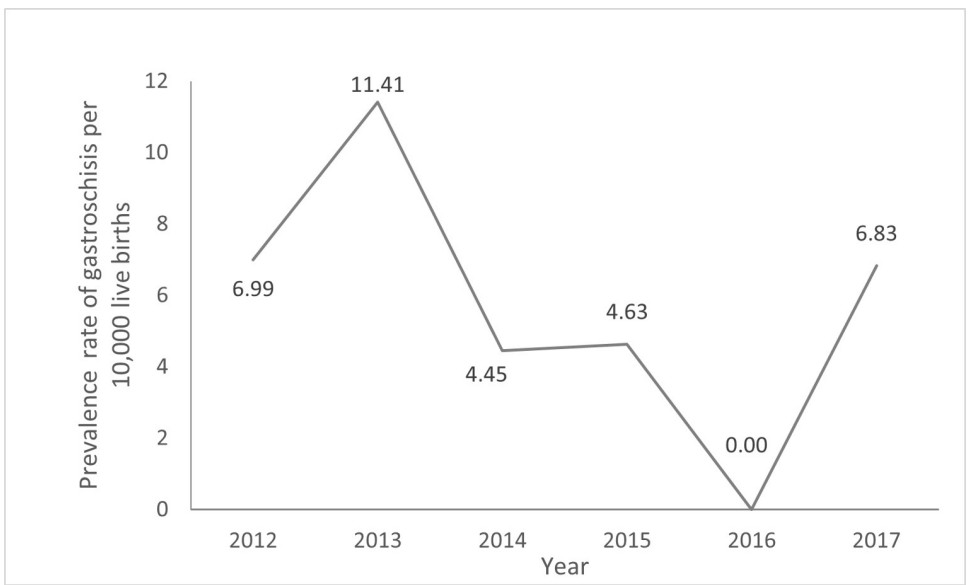

**Fig 1. Prevalence rate of gastroschisis in Foz do Iguassu from 2012 to 2017.**

## Results

A total of 15 gastroschisis cases were recorded to 26,182 births from 2012 to 2017 in Foz do Iguassu. The prevalence rate average was 5.73/10,000 live birth in the period (Fig 1).

Fig 2A shows the map with urban census sectors of Foz do Iguassu. It is observed that PTL is located in the northern region. Spatial distribution of gastroschisis shows highest in north (minimum = 0; maximum = 20; average = 0.51; standard deviation = 2.54). Univariate Global Moran´s I analysis did not show spatial dependency of gastroschisis prevalence rate (Moran´s I = 0.006; p = 0.319) (Fig 2B).

LISA did not identify significant High-High or Low-Low clusters (Fig 3A). However, Getis-Ord statistics identified 29 census sectors with high type (hot spot) and 233 census sectors with low type (cold spot) clusters of gastroschisis rates (Fig 3B).

The spatial scan statistic identified a significant region with low risk for gastroschisis (Circle D, OR = 0; p = 0.035) in the southern region of the city (Fig 4).

In simple logistic regression, independent variables RAP (OR 7.07; CI 95% 1.57–31.89; p = 0.010), RAAP (OR 0.23; CI 95% 0,06–0,85; p = 0,027), population with income above 5 minimum wages (OR 0.23; CI 95% 0,06–0,86; p = 0,029) and proximity to PTL (OR 5,96; CI 95% 2,05–17,37; p = 0,001) were associated with gastroschisis prevalence. Multiple logistic regression analysis showed only the proximity to PTL (OR 3,47; CI 95% 1,11–10,79; p = 0,031) remained associated (Table 1).

## Discussion

This is the first study to investigate the association of socio-environmental factors and gastroschisis with possible impact of PTL. No global spatial dependency was observed in the distribution of gastroschisis. However, spatial scan statistic showed low risk for gastroschisis in areas opposite to PTL. In addition, multiple logistic regression showed high spot sectors with higher chance of being close to PTL.

Low frequency EMF has been shown to not cause DNA breaks [4]. However, some studies suggest it may react to cell membrane [6, 7, 26]. Although the 50/60 Hz seems to not directly

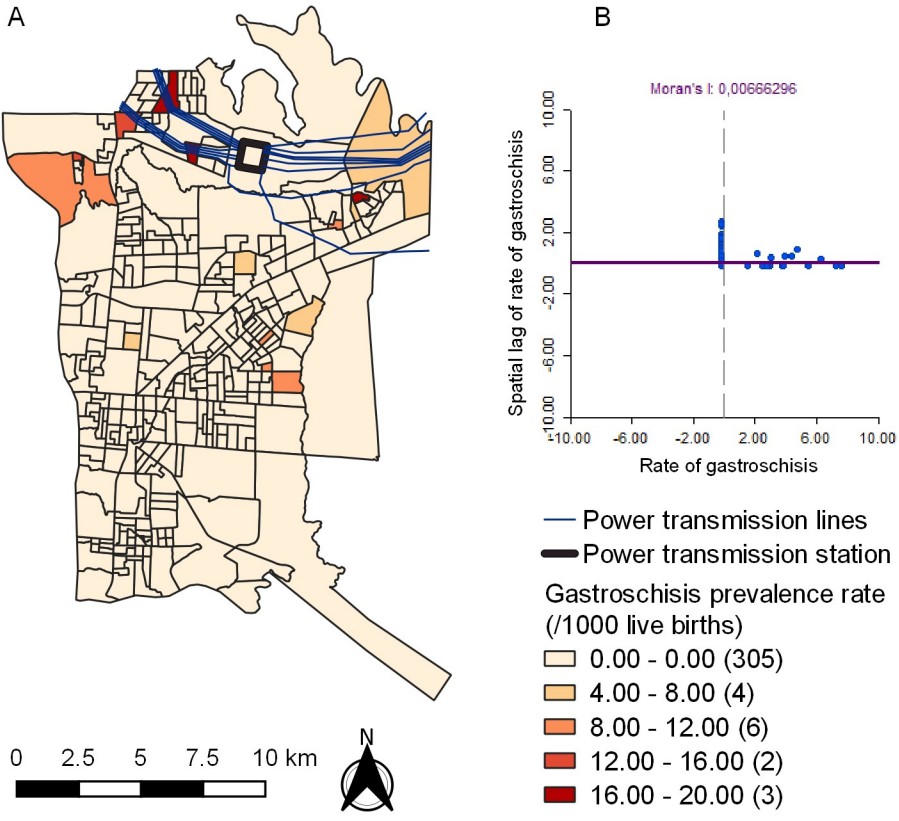

**Fig 2.** Spatial distribution of the prevalence rate of gastroschisis (A) and Univariate Global Moran´s Index Scatter Plot (B) in Foz do Iguassu from 2012 to 2017. Republished from Shapefile maps with SIRGAS2000 projection / UTM zone 21S under a CC BY license, with permission from Brazilian Institute of Geography and Statistics, original copyright 2020.

cause genotoxic effects, the increase in free radicals may lead to genome instability, micronuclei formation and DNA repair dysfunction [6, 7, 26].

Most studies performed to establish a relation between EMF and diseases involve the incidence of cancer. The association between proximity to PTL and childhood leukemia, brain tumors and breast cancer has been described [27–30]. In relation pregnancy risk, maternal residence may be related to abortion, congenital anomaly and prematurity [8–10]. In general, studies consider distances of ≤50, ≤100 and / or ≤500 meters from PTL as exposure factor.

Spatial epidemiology provides early risk information and timetable public health interventions in these areas. Simple logistic regression model showed incomes above 5 minimum wages was a protective factor against the prevalence of gastroschisis. In fact, income is reported as a determinant social determinant of health [31]. In particular, congenital anomalies provide information about nutrition condition and risk exposure to health population.

## Limitation

The effects of environmental exposure on congenital anomalies present many challenges. Congenital anomalies are less common and difficult to obtain high statistical power to stablish standards. The association between environmental exposure and congenital anomalies is also

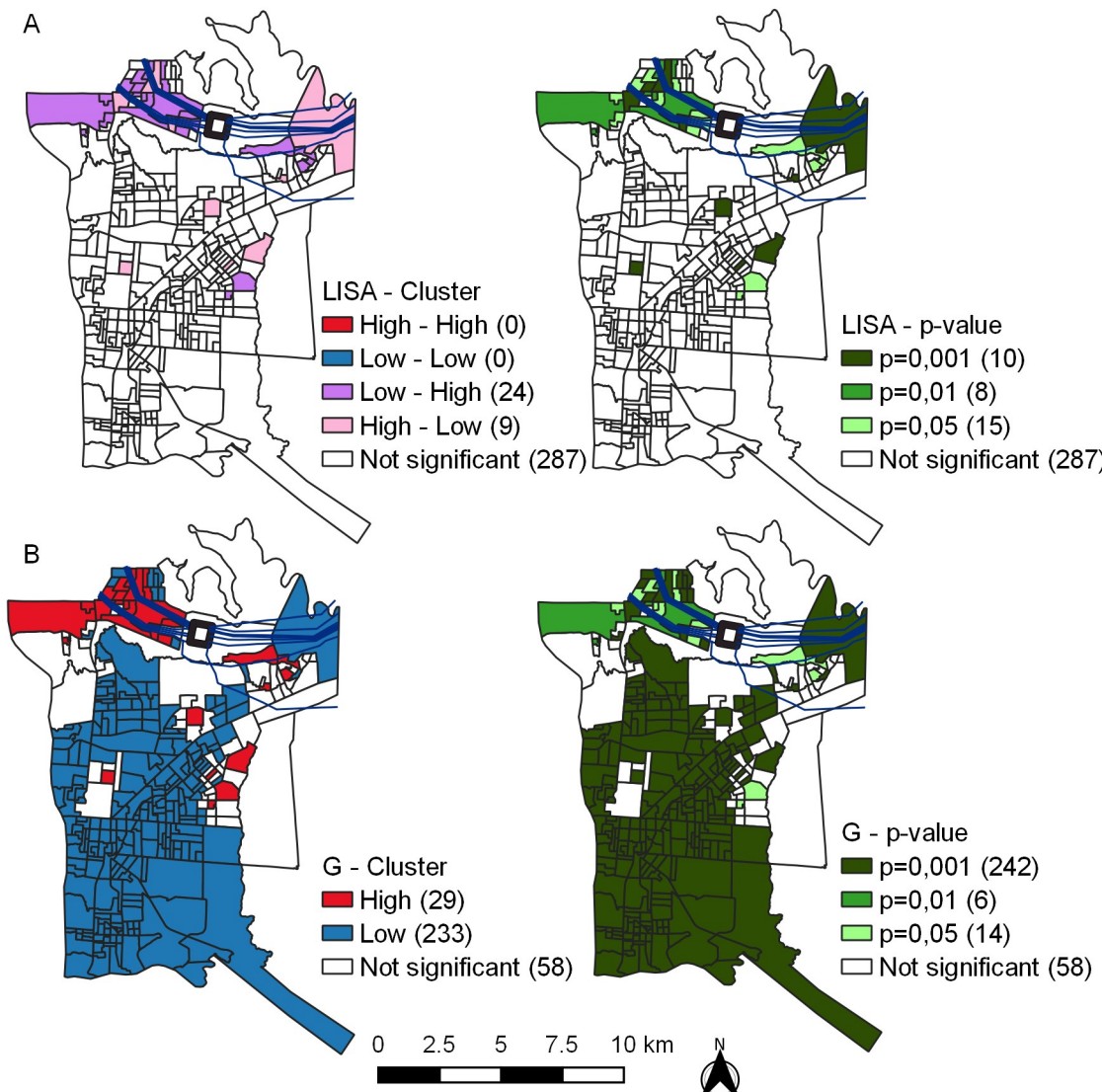

A - Local Index of Spatial Association for prevalence rate of gastrosquise
B - Gettis-Ord statistics for prevalence rate of gastrosquise

**Fig 3.** Local indicator of spatial association analysis (A) and Getis-Ord statistics of prevalence rate of gastroschisis(B) in Foz do Iguassu from 2012 to 2017. Republished from Shapefile maps with SIRGAS2000 projection/UTM zone 21S under a CC BY license, with permission from Brazilian Institute of Geography and Statistics, original copyright 2020.

influenced by other individual behavior or labor health quality. We were unable to determine EMF intensity in the exposed areas.

## Conclusion

No global spatial dependency was observed in the distribution of gastroschisis in Foz do Iguassu. However, census sectors with anomaly cases had a higher chance of being close to

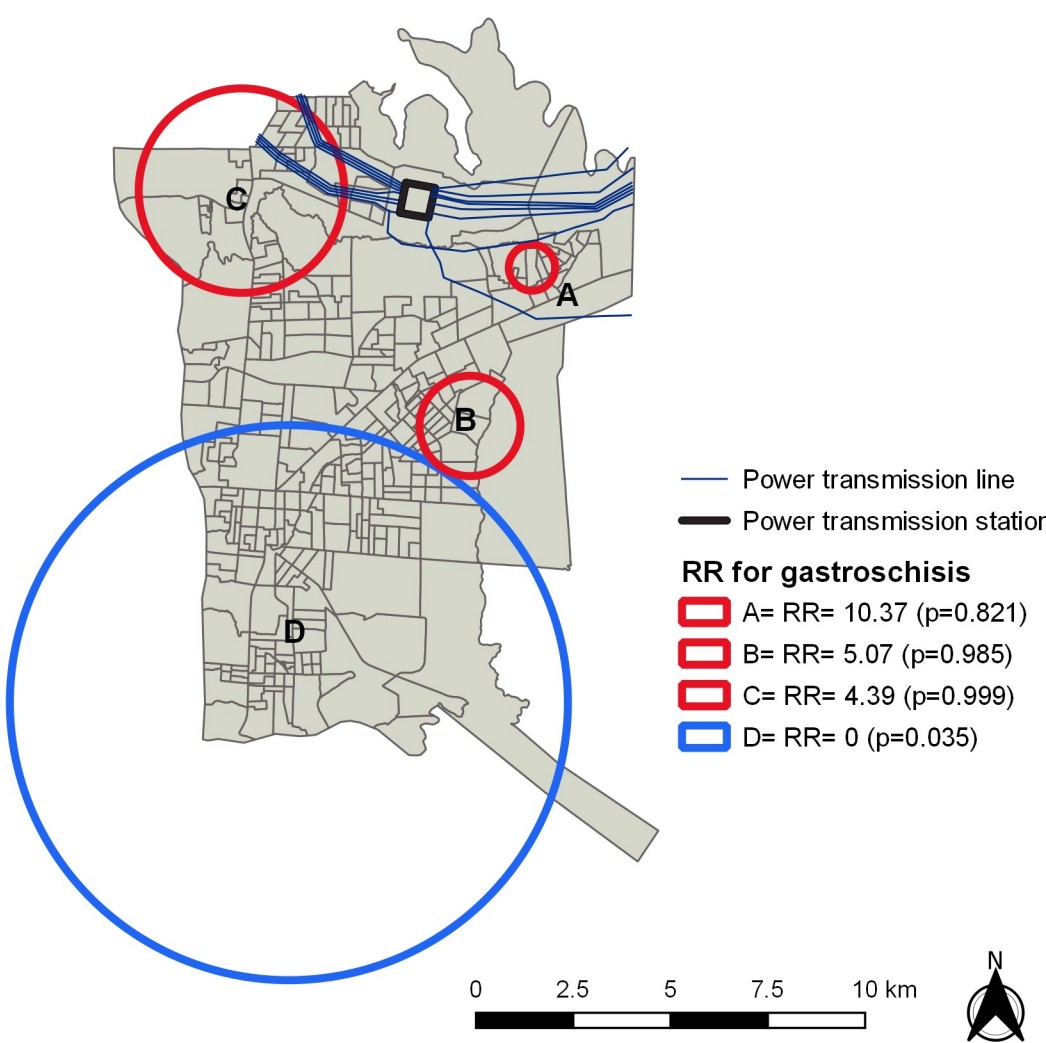

**Fig 4. Relative Risk areas for gastroschisis in Foz do Iguassu from 2012 to 2017.** Republished from Shapefile maps with SIRGAS2000 projection/UTM zone 21S under a CC BY license, with permission from Brazilian Institute of Geography and Statistics, original copyright 2020.

**Table 1. Simple and Multiple logistic regression analysis of socioenvironmental factors associated with gastroschisis prevalence in Foz do Iguassu from 2012 to 2017.**

| | Simple Regression | | | Multiple Regression | | |
|---|---|---|---|---|---|---|
| | OR | CI95% | p-value | OR | CI95% | p-value |
| Rate of adolescent parturients | 7,07 | 1,57–31,89 | 0,010 | 3,38 | 0,63–18,07 | 0,153 |
| Rate of parturient over 35 years | 0,23 | 0,06–0,85 | 0,027 | 0,38 | 0,10–1,48 | 0,167 |
| Population without income | 0,96 | 0,34–2,72 | 0,947 | - | - | - |
| Population with income above 5 minimum wages | 0,23 | 0,06–0,86 | 0,029 | 0,77 | 0,18–3,35 | 0,736 |
| Rate of Late Prenatal | 1,40 | 0,49–3,96 | 0,524 | - | - | - |
| Proximity to PLT | 5,96 | 2,05–17,37 | 0,001 | 3,47 | 1,11–10,79 | 0,031 |

PTL despite no causality between EMF and gastroschisis could be determined in this study. Spatial observation of the distribution cases can contribute to the management of health care for pregnant and newborns in more susceptible areas.

## Supporting information

**S1 Data.**
(XLSX)

## Acknowledgments

The authors thank the Professor Marcos Augusto Moraes Arcoverde for the statistical consultancy.

## Author Contributions

**Conceptualization:** Suzana de Souza, Cezar Rangel Pestana.

**Data curation:** Suzana de Souza.

**Formal analysis:** Suzana de Souza, Oscar Kenji Nihei.

**Investigation:** Suzana de Souza, Oscar Kenji Nihei.

**Methodology:** Suzana de Souza, Oscar Kenji Nihei.

**Project administration:** Cezar Rangel Pestana.

**Software:** Oscar Kenji Nihei.

**Visualization:** Suzana de Souza.

**Writing – original draft:** Suzana de Souza.

**Writing – review & editing:** Suzana de Souza, Oscar Kenji Nihei, Cezar Rangel Pestana.

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
