## [Decision Letter · Decision Letter 0]

10 Dec 2020

PONE-D-20-30903

High incidence of gastroschisis in Brazilian triple side border: a socioenvironmental spatial analysis

PLOS ONE

Dear Dr. de Souza,

Thank you for submitting your manuscript to PLOS ONE. After careful consideration, we feel that it has merit but does not fully meet PLOS ONE’s publication criteria as it currently stands. Therefore, we invite you to submit a revised version of the manuscript that addresses the points raised during the review process.

I know have the review of this work. Although the area and topic are certainly of interest there are suggestions of extensive revisions on all aspects of the submitted work.

We look forward to receiving your revised manuscript.

Kind regards,

JJ Cray Jr., Ph.D.

Academic Editor

PLOS ONE

Journal Requirements:

2. In the methods section, please provide additional information regarding how study variables were extracted from the database for analysis. Please ensure that you have described this in sufficient detail to allow your work to be replicated.

3.We note that you have indicated that data from this study are available upon request. PLOS only allows data to be available upon request if there are legal or ethical restrictions on sharing data publicly. For information on unacceptable data access restrictions, please see http://journals.plos.org/plosone/s/data-availability#loc-unacceptable-data-access-restrictions.

4.We note that [Figure(s) 2, 3 and 4] in your submission contain map images which may be copyrighted. All PLOS content is published under the Creative Commons Attribution License (CC BY 4.0), which means that the manuscript, images, and Supporting Information files will be freely available online, and any third party is permitted to access, download, copy, distribute, and use these materials in any way, even commercially, with proper attribution. For these reasons, we cannot publish previously copyrighted maps or satellite images created using proprietary data, such as Google software (Google Maps, Street View, and Earth). For more information, see our copyright guidelines: http://journals.plos.org/plosone/s/licenses-and-copyright.

1.    You may seek permission from the original copyright holder of Figure(s) [2, 3 and 4] to publish the content specifically under the CC BY 4.0 license. 

Reviewers' comments:

Reviewer's Responses to Questions

**Comments to the Author**

1. Is the manuscript technically sound, and do the data support the conclusions?

Reviewer #1: Yes

2. Has the statistical analysis been performed appropriately and rigorously? 

Reviewer #1: Yes

3. Have the authors made all data underlying the findings in their manuscript fully available?

Reviewer #1: No

4. Is the manuscript presented in an intelligible fashion and written in standard English?

Reviewer #1: No

5. Review Comments to the Author

Reviewer #1: Dear Dr. JJ Cray

I am very thankful for the invitation to review the manuscript for PLOS ONE entitled "High incidence of gastroschisis in Brazilian triple side border: a socioenvironmental spatial analysis" (PONE-D-20-30903).

I did use the very best of my knowledge to help you decide and the authors to improve their manuscript.

Overall, the authors did an excellent research regarding the exposure of magnetic fields, and their association with live births with gastroschisis in Foz do Iguassu, a city located in Parana state – Brazil, which hosts one of the world's biggest hydroelectric dams.

They respected the manuscript organization present in the instruction to authors on the PLOS website. It has organized in Title, a non-structured Abstract, Background, Material and Methods, Results, Discussion, Limitation, Conclusion, Acknowledgments, References and presents the Result's figures at the end.

Observations regarding each manuscript section are below.

1) Title

It is specific, descriptive, and draws attention to the present question.

No recommendations in this section.

2) Abstract

It is a non-structured abstract that describes the study's primary objective, explains the method's principal points, and shows the main results and conclusion.

No recommendations in this section.

3) Background

The authors summarized the gastroschisis problem very well; nevertheless, PREVALENCE is the usual term in the medical literature when referring to a frequency measure of any congenital disease, not INCIDENCE.[1, 2]

I recommend altering the term INCIDENCE for PREVALENCE in this and all the other sections and figures.

1. Hook EB. Incidence and prevalence as measures of the frequency of birth defects. American journal of epidemiology. 1982;116(5):743-7.

2. Mason CA, Kirby RS, Sever LE, Langlois PH. Prevalence is the preferred measure of frequency of birth defects. Birth defects research Part A, Clinical and molecular teratology. 2005;73(10):690-2. Epub 2005/10/22. doi: 10.1002/bdra.20211. PubMed PMID: 16240384.

4) Materials and Methods

A) Study design, setting, and population

Line 74 - The term newborn covers the population of live births and stillbirths. As SINASC uses only the live births population, it would be better to replace the term to avoid confusion.

Line 74 – Is not the six years period (2012 – 2017) short for this analysis? Publications numbers 27 – 30 in the REFERENCES section present a study period that varies from 7 to 33 years.

The alteration of the term and the explanation for the six-year period are the recommendations in this section.

B) Data sources and study variables

All the SINASC database is available on the DATASUS website (www.datasus.gov.br), including the MICRODATA used in this research (http://www2.datasus.gov.br/DATASUS/index.php?area=0901&item=1&acao=28&pad=31655).

My recommendation in this section is to:

- Alter the statement in "Data Availability": NO – to YES, and refer to the DATASUS WEBSITE.

- Alter the statement that "Data cannot be shared publicly" because they already are public. In "Describe where the data may be found……appropriate details."

- The technical details should be expanded and clarified to ensure that readers understand precisely the steps made. It is not only to export the SINASC data to an Excel spreadsheet; TABWIN should be used first.

- The technical details should be expanded and clarified to ensure that readers precisely understand the steps to obtain all the SINASC and IBGE data.

C) Power Transmission Lines

No recommendations in this section.

D) Data analysis

The alteration to PREVALENCE instead of the term INCIDENCE as recommended before.

E) Global Moran's Index

No recommendations in this section.

F) Local indicator of spatial association analysis

No recommendations in this section.

G) Getis-Ord statistics

No recommendations in this section.

H) Spatial scan statistic

No recommendations in this section.

I) Logistic regression

Line 208: The authors used a distance of 850 meters between the centroid of the census sector to the closest point in PTL as an independent variable. No other paper in the REFERENCE section uses this distance (Ref 10: 600m, Ref 29: <200m, 200-600m, >600m, Ref 30: 500m), and in the DISCUSSION section line 272, the authors also describes the distances. So, the question is, why did the authors use 850m as standard?

The authors should clarify and expand the technical details to a better understanding.

Line 220: The authors could create another subsection entitle: Ethics Review to present the Ethics data from Plataforma Brasil, apart from Logistic regression.

5) Results

Line 225: (15 / 26,182) X 10,000 = 5.73 (PREVALENCE rate 2012 - 2017) not 5.75

Line 266, 228 & 241: Change the term Incidence for Prevalence

FIG 1, 2, 3 & Table 1: Change the term Incidence for Prevalence

6) Discussion

While the study appears to be sound, the language is unclear, making it difficult to follow. Please advise the authors to work with a writing coach or copyeditor to improve the text's flow and readability, principally in the Discussion's first paragraph.

7) Limitation

No recommendations in this section.

8) Conclusion

No recommendations in this section.

9) Acknowledgments

No recommendations in this section.

10) References

No recommendations in this section.

Overall, the manuscript's idea is outstanding. A major revision will be required, and an English revision from a Native speaker or a writing editing service. Some points must be better explained to clarify and give a better understanding to the readers.

Sincerely

Mauricio Giusti Calderon M.D, Ph.D

6. PLOS authors have the option to publish the peer review history of their article (what does this mean?). If published, this will include your full peer review and any attached files.

Reviewer #1: **Yes: **MAURICIO GIUSTI CALDERON, M.D ,Ph.D

---

## [Author Response · Author response to Decision Letter 0]

5 Feb 2021

RESPONSES TO THE REVIEWERS

PONE-D-20-30903

High incidence of gastroschisis in Brazilian triple side border: a socioenvironmental spatial analysis

Dear Reviewers, 

We would like to thank you for the careful review and contribution to the paper. Journal requirements and reviewers´ comments are addressed in the responses bellow. Also, a revised marked version is also presented with changes highlighted in yellow.

Yours sincerely, 

Suzana de Souza

Journal Requirements:

2. In the methods section, please provide additional information regarding how study variables were extracted from the database for analysis. Please ensure that you have described this in sufficient detail to allow your work to be replicated.

Authors’ answer: We have provided more information regarding study variables. Please see line 79 in Data sources and study variables subsection:

“The variables obtained from SINASC were:

• Type of congenital anomaly (Gastroschisis (Q79.3, according to International Classification of Diseases – ICD));

• Parturients age (Presented at SINASC as a continuous quantitative variable; in this research was categorized as adolescent (up to 19 years old), adult (20 to 34 years old) and advanced age (over 35 years old)).

• Prenatal start period (Presented at SINASC as a continuous quantitative variable; in this research was categorized as early prenatal care (beginning in the first semester of pregnancy) and late prenatal care (beginning after the first semester of pregnancy)).”

3. We note that you have indicated that data from this study are available upon request. PLOS only allows data to be available upon request if there are legal or ethical restrictions on sharing data publicly.

Authors’ answer: Data set are not subject to legal or ethical restriction. Minimal anonymized was upload to protect patient information. 

4. We note that [Figure(s) 2, 3 and 4] in your submission contain map images which may be copyrighted. All PLOS content is published under the Creative Commons Attribution License (CC BY 4.0), which means that the manuscript, images, and Supporting Information files will be freely available online, and any third party is permitted to access, download, copy, distribute, and use these materials in any way, even commercially, with proper attribution. For these reasons, we cannot publish previously copyrighted maps or satellite images created using proprietary data.

Authors’ answer: Brazilian Institute of Geography and Statistics (IBGE) declared all extracted data (shapefile) is public and can be freely reproduced with source indicated. We have indicate source in each image. Written permission is upload. Please also see lines 230, 236 and 243 in Results section: “Source: Brazilian Institute of Geography and Statistics, 2010. SIRGAS2000 projection / UTM zone 21S.”

Reviewers' comments:

3) Background

The authors summarized the gastroschisis problem very well; nevertheless, PREVALENCE is the usual term in the medical literature when referring to a frequency measure of any congenital disease, not INCIDENCE.

I recommend altering the term INCIDENCE for PREVALENCE in this and all the other sections and figures.

Authors’ answer: We have changed the term INCIDENCE to PREVALENCE in these passages. Please see lines 44, 122, 219 and 272. 

4) Materials and Methods

A) Study design, setting, and population

Line 74 - The term newborn covers the population of live births and stillbirths. As SINASC uses only the live births population, it would be better to replace the term to avoid confusion.

Authors’ answer: We have now used only the term "live births". 

Line 74 – Is not the six years period (2012 – 2017) short for this analysis? Publications numbers 27 – 30 in the REFERENCES section present a study period that varies from 7 to 33 years.

The alteration of the term and the explanation for the six-year period are the recommendations in this section.

Authors’ answer: Live Birth Information System (SINASC) was created in 1990 but its implementation occurred only gradually in all Federation Units. We chose this period to collect more recent and consistent data to minimize bias in the study.

D) Data analysis

The alteration to PREVALENCE instead of the term INCIDENCE as recommended before.

Authors’ answer: We have changed the term INCIDENCE to PREVALENCE in these passages.

I) Logistic regression

Line 208: The authors used a distance of 850 meters between the centroid of the census sector to the closest point in PTL as an independent variable. No other paper in the REFERENCE section uses this distance (Ref 10: 600m, Ref 29: <200m, 200-600m, >600m, Ref 30: 500m), and in the DISCUSSION section line 272, the authors also describes the distances. So, the question is, why did the authors use 850m as standard?

The authors should clarify and expand the technical details to a better understanding.

Authors’ answer:, Unlike other studies, our distance analysis was based on the centroid of each census sector. After empirical tests, we found that the best distance to cover all census sectors near the power transmission lines was 850 meters. We added a sentence in order to better explain this choice. Please see line 202 in Logistic regression subsection: “After empirical tests, the distance that included all census sector close to PTL was 850 meters”.

Line 220: The authors could create another subsection entitle: Ethics Review to present the Ethics data from Plataforma Brasil, apart from Logistic regression.

Authors’ answer: We created a new subsection according to the suggestion. Please see line 213.

5) Results

Line 225: (15 / 26,182) X 10,000 = 5.73 (PREVALENCE rate 2012 - 2017) not 5.75

Authors’ answer: We fixed the error in the sentence.

Line 266, 228 & 241: Change the term Incidence for Prevalence

FIG 1, 2, 3 & Table 1: Change the term Incidence for Prevalence

Authors’ answer: We have changed the term INCIDENCE to PREVALENCE in all sections of the manuscript.

6) Discussion

While the study appears to be sound, the language is unclear, making it difficult to follow. Please advise the authors to work with a writing coach or copyeditor to improve the text's flow and readability, principally in the Discussion's first paragraph.

Authors’ answer: The manuscript was extensively revised. A new version is presented after English grammar and language improvements.

---

## [Decision Letter · Decision Letter 1]

16 Feb 2021

High prevalence of gastroschisis in Brazilian triple side border: a socioenvironmental spatial analysis

PONE-D-20-30903R1

Dear Dr. de Souza,

We’re pleased to inform you that your manuscript has been judged scientifically suitable for publication and will be formally accepted for publication once it meets all outstanding technical requirements.

Kind regards,

JJ Cray Jr., Ph.D.

Academic Editor

PLOS ONE

Additional Editor Comments (optional):

Reviewers' comments:

Reviewer's Responses to Questions

**Comments to the Author**

1. If the authors have adequately addressed your comments raised in a previous round of review and you feel that this manuscript is now acceptable for publication, you may indicate that here to bypass the “Comments to the Author” section, enter your conflict of interest statement in the “Confidential to Editor” section, and submit your "Accept" recommendation.

Reviewer #1: All comments have been addressed

2. Is the manuscript technically sound, and do the data support the conclusions?

Reviewer #1: Yes

3. Has the statistical analysis been performed appropriately and rigorously? 

Reviewer #1: Yes

4. Have the authors made all data underlying the findings in their manuscript fully available?

Reviewer #1: Yes

5. Is the manuscript presented in an intelligible fashion and written in standard English?

Reviewer #1: Yes

6. Review Comments to the Author

Reviewer #1: Congratulations.

All points raised were duly explained or corrected in a scientifically appropriate manner.

SINASC is a powerful tool, but little used for epidemiological studies of congenital malformations in Brazil, I hope you will follow this research line not only for gastroschisis, but also for other pathologies.

7. PLOS authors have the option to publish the peer review history of their article (what does this mean?). If published, this will include your full peer review and any attached files.

Reviewer #1: **Yes: **MAURICIO GIUSTI CALDERON

---

## [Editor Report · Acceptance letter]

18 Feb 2021

PONE-D-20-30903R1 

High prevalence of gastroschisis in Brazilian triple side border: a socioenvironmental spatial analysis 

Dear Dr. de Souza:

I'm pleased to inform you that your manuscript has been deemed suitable for publication in PLOS ONE. Congratulations! Your manuscript is now with our production department. 

Kind regards, 

on behalf of

Dr. JJ Cray Jr. 

Academic Editor

PLOS ONE